# Biogas Slurry Significantly Improved Degraded Farmland Soil Quality and Promoted *Capsicum* spp. Production

**DOI:** 10.3390/plants13020265

**Published:** 2024-01-17

**Authors:** Zichen Wang, Isaac A. Sanusi, Jidong Wang, Xiaomei Ye, Evariste Gueguim Kana, Ademola O. Olaniran

**Affiliations:** 1Institute of Agricultural Resources and Environment, Jiangsu Academy of Agricultural Sciences, Nanjing 210014, China; 20160028@jaas.ac.cn; 2Discipline of Microbiology, School of Life Sciences, University of KwaZulu-Natal, Pietermaritzburg 4000, South Africa; sanusi_isaac@yahoo.com (I.A.S.); olanirana@ukzn.ac.za (A.O.O.); 3Key Laboratory of Crop and Livestock Integrated Farming, Ministry of Agriculture and Rural Affairs, Nanjing 210014, China; yexiaomei610@126.com; 4Key Laboratory of Saline-Alkali Soil Improvement and Utilization (Coastal Saline-Alkali Lands), Ministry of Agriculture and Rural Affairs, Nanjing 210014, China; 5Liuhe Observation and Experimental Station of National Agricultural Environment, Nanjing 210014, China

**Keywords:** biogas slurry, plant–environment interaction, soil fertility increase and maintenance, liquid digestate, salt-affected soil, improvement, eco-restoration

## Abstract

This study reports on the effects of pretreated biogas slurry on degraded farm soil properties, microflora and the production of *Capsicum* spp. The responses of soil properties, microorganisms and *Capsicum* spp. production to biogas slurry pretreated soil were determined. The biogas slurry pretreatment of degraded soil increases the total nitrogen (0.15–0.32 g/kg), total phosphorus (0.13–0.75 g/kg), available phosphorus (102.62–190.68 mg/kg), available potassium (78.94–140.31 mg/kg), organic carbon content (0.67–3.32 g/kg) and pH value of the soil, while the population, diversity and distribution of soil bacteria and fungi were significantly affected. Interestingly, soil ammonium nitrogen, soil pH and soil nitrate nitrogen were highly correlated with the population of bacteria and fungi present in the pretreated soil. The soil with biogas slurry pretreatment of 495 m^3^/hm^2^ favored the seedling survival rate, flowering rate and fruit-bearing rate of *Capsicum* spp. and significantly reduced the rate of rigid seedlings. In this study, the application of 495 m^3^/hm^2^ biogas slurry to pretreat degraded soil has achieved the multiple goals of biogas slurry valorization, soil biofertilization and preventing and controlling plant diseases caused by soil-borne pathogenic microorganisms. These findings are of significant importance for the safe and environmentally friendly application of biogas slurry for soil pretreatment.

## 1. Introduction

The extensive application of acidic or physiologically acidic fertilizers, pesticides and high-intensity single planting modes in protected vegetable cultivation can easily cause soil quality degradation, soil-borne plant diseases, soil acidification, secondary salinization, nutrient imbalance and other continuous cropping obstacles [1]. These usually result in a sharp decline in vegetable yield and sustainable vegetable cultivation, seriously affecting the economic income of farmers. It is reported that the losses caused by pathogenic bacteria in the vegetable industry around the world are more than USD 1 billion every year [2]. Research on the effective methods of preventing and controlling soil degradation as well as soil-borne plant diseases in protected lands has become imperative and important for promoting sustainable global agricultural development.

Soil-borne plant diseases are plant diseases caused by pathogenic fungi, bacteria, nematodes and viruses in the soil. The microbes can proliferate when the soil conditions are suitable, infecting the roots and stems of crops [3]. Among them, root rot disease in protected land vegetables caused by pathogenic fungi is more serious, and after entering the planting period, there will be a large number of dead plants. The initial symptoms include single plants becoming stiff, wilting, and dying. Later on, it may even lead to the death of all greenhouse plants, and in severe cases, the yield can be reduced by more than 60%. Additionally, soil acidification provides a suitable living environment for the growth of pathogenic fungi [4]. The accumulation of phenolic acid allelochemicals in soil affects the permeability of root cell membranes, which is the main reason for the autotoxic effect of continuous cropping on vegetables, and this provides pathogenic fungi with a carbon source and energy for growth [4]. Scholars from Japan, the Netherlands, the United States, and China have made efforts to mitigate soil-borne diseases such as Reductive Soil Disinfestation (RSD). The RSD method has proven to be a very strong method of “curing” infection by soil-borne pathogens [5,6,7]. The core method is to apply a large amount of easily decomposed organic materials, irrigate and mulch to prevent air from diffusing into the soil, creating a strong soil reduction state in a short time, thus killing soil-borne pathogens.

As a by-product of biogas production, biogas slurry is weakly alkaline, and it can be used to alleviate soil acidification. Additionally, biogas slurry is rich in organic carbon and nutrient elements, which can provide rich carbon sources and energy for microbial growth. Hence, the application of biogas slurry to soil could significantly influence the population, community structure and activity of soil microorganisms [8]. Biogas slurry treatment has been found to increase the population of bacteria such as *Pseudomonas fluorescens* and *Trichoderma*, fungi in the rhizosphere soil, as well as improve the diversity index of soil bacteria and fungi [9]. In another related study, Sui et al. [10] showed that the population of soil actinomycetes and fungi increased by 72.4% and 61.6%, respectively, while the number of soil bacteria decreased by 18.4% when biogas slurry drip irrigation was used in a solar greenhouse. Moreover, biogas slurry has a certain inhibitory effect on a variety of crop-pathogenic fungi [11]. Li et al. [12] reported a significant inhibitory effect on *Rhizoctonia solani*, *Fusarium nivale*, *Fusarium oxysporum* and *Fusarium solani*, with an inhibition rate of 70%, 40%, 68% and 70%, respectively, when biogas slurry was used for the treatment of farmland. Similarly, the biogas slurry of 21 different large-scale biogas projects in Jiangsu Province of China had different degrees of inhibitory effect on the growth of strawberry *Fusarium* wilt when applied to farmland [13]. In addition, the stage of obtaining the biogas slurry had a significant effect on its antibacterial effect [13]. Despite the fact that there have been different applications of biogas slurry on farmland as a potential biofertilizer and pesticide, there is a scarcity of knowledge on the use of biogas slurry to pretreat degraded protected soil. Similarly, there is a dearth of information on preventing and controlling the soil-borne diseases of protected farmland using biogas slurry. Knowledge concerning multiple potential usages of biogas slurry, the impact of biogas slurry on soil and its operating mechanism provides a theoretical basis and technical approach to biogas slurry farmland applications, thereby improving the resource utilization of biogas slurry.

Furthermore, studies have shown that the application of biogas slurry can improve the physical and chemical properties of farmland while effectively valorizing biogas slurry [14,15,16]. The treatment of farmland with biogas slurry has a direct positive effect on improving soil organic matter, soil structure and fertility [17,18]. Overall, the pretreatment of farmland with biogas slurry has multiple benefits for the soil and cultivated crops. Thus, it is imperative to study the dual benefits of biogas slurry on soil physiochemical properties and the growth performance of cultivated crops.

Hence, the present study assessed the application of biogas slurry to pretreat the degraded protected soil to achieve the dual goals of preventing and controlling plant diseases caused by soil-borne pathogens in the protected farmland. Additionally, the efficiency of biogas slurry absorption per unit area of farmland was evaluated through field experiments. The changes in soil properties, soil microbial community and the growth of *Capsicum* spp. following the application of biogas slurry was also analyzed, based on the RSD method.

## 2. Results

### 2.1. Effect of Biogas Slurry Pretreatment on Soil Properties

The impact of biogas slurry on soil properties is shown in Figure 1. After 20 days of biogas slurry treatment of the soil, compared with CK treatment, LBS, HBS, and LBSM treatment increased TN content by 25.3%, 11.7%, and 19.0%, respectively (Figure 1A). The HBS, LBSM and HBSM treatments significantly increased the content of NH_4_^+^-N by 24.6%, 33.1% and 86.8%, respectively, while LBS significantly reduced the content of ammonium nitrogen by 14.2% (Figure 1B). On the other hand, all the treatments significantly reduced NO_3_^−^-N content with LBS, HBS, LBSM, HBSM and WM treatments reducing NO_3_^−^-N content by 47.8%, 52.2%, 51.3%, 55.9% and 84%, respectively (Figure 1C).

Moreover, the LBS, HBS, LBSM and HBSM treatments increased the TP content by 79.2%, 55.8%, 65.2% and 14.2%. LBS, HBS and LBSM treatments increased the TP content significantly (*p* < 0.05) (Figure 1D). Similarly, the LBS, HBS, LBSM and HBSM treatments increased the content of available phosphorus and potassium in the treated soil. The content of available phosphorus increased by 1378.5%, 1355.9%, 778.6% and 741.9% (Figure 1E), while the content of available potassium increased by 87.7%, 84.8%, 61.0% and 108.4%, respectively (Figure 1F).

Likewise, the LBS, HBS, LBSM and HBSM-treated soil increased in soil organic carbon content by 38.7%, 27.4%, 40.3% and 8.1% (organic carbon content in LBS, HBS and LBSM treatments was observed to increase significantly (*p* < 0.05)) (Figure 1G). Additionally, the pH value in the HBS, LBSM and HBSM-treated soil increased significantly by 0.8%, 2.2% and 2.5%, respectively (Figure 1H).

### 2.2. Effect of Biogas Slurry Pretreatment on Soil Microorganisms

#### 2.2.1. Soil Culturable Microorganisms

Shown in Figure 2 is the impact of different pretreatment measures on culturable bacteria, fungi, actinomycetes (a group of Gram-positive bacteria, different from bacteria in terms of cellular characteristics and metabolic functions, actinomycetes can produce various secondary metabolites such as antibiotics and play an important role in controlling harmful microorganisms in soil) and genus *Fusarium* (all culturable *Fusarium* species, including the pathogenic fungus causing *Capsicum* spp. root rot). The results obtained showed that the soil pretreatment for 15 days significantly reduced the number of culturable bacteria in the soil (Figure 2A). There was an increase in the population of culturable bacteria in the soil after the biogas slurry application when compared with the control (CK treatment), while the difference in the population of culturable bacteria between treatments decreases with the increase in pretreatment time. After the cultivation period of 20 days, the number of culturable bacteria in LBSM treatment reduced compared to CK treatment (33.3%). Similarly, the fungi load of culturable fungi in the soil increased first and then decreased after 20 days of pretreatment (Figure 2B). On the other hand, comparing LBS and HBS treatment to CK treatment, LBS and HBS treatment increased the number of culturable fungi by 130.1% and 30.1%, while LBSM and HBSM treatment resulted in a decrease in the fungi load of culturable fungi by 41.0% and 13.0%, respectively, without observable significant difference (*p* < 0.05).

Moreover, in Figure 2C, a gradual increase in the actinomyces population was observed, and, although the cultivation period was extended, there was no substantial effect on the population of actinomyces. On the 20th day, the number of actinomycetes decreased by 31.1%, 54.5%, 42.6% and 52.9% for LBS, HBS, LBSM and HBSM treatment, respectively. The reduction in actinomyces was found to be significant within treatment and in comparison to control (*p* < 0.05).

Furthermore, biogas slurry pretreatment of farmland reduced the population of cultivable *Fusarium* in the soil (Figure 2D). The LBS-treated soil showed *Fusarium* population initially increased (1.19 × 10^2^ cfu/g at day 5, 7.90 × 10^2^ cfu/g at day 10, and 1.02 × 10^3^ cfu/g at day 15) and then decreased (1.20 × 10^2^ cfu/g at day 20) as the treatment period extended. LBSM treatment showed a decreasing trend in the *Fusarium* population, while HBS and HBSM-treated soil showed an initial decrease in the *Fusarium* population, then increased to a later decrease. When these are compared to CK treatment, LBS, HBS, LBSM and HBSM treatment significantly decreased the number of cultivable *Fusarium* in soil by 16.4%, 48.4%, 77.9% and 70.5%, respectively, after the 15th day of treatment. On the 20th day, the population of cultivable *Fusarium* in each treatment decreased less than that in CK treatment, but there was no significant difference in the population of cultivable *Fusarium* between LBS, LBSM and CK treatment.

#### 2.2.2. Changes in Soil Microbial Community Diversity

Figure 3 shows the OTU cluster analysis and annotation results of high-throughput sequencing of bacterial 16S and fungal ITS in different soil treatments. Compared with CK treatment, the number of soil bacterial OTU in BS (LBS + HBS) and BSM (LBSM + HBSM) treatment decreased by 12.9% and 49.3%, respectively. Moreover, the population of fungi OTU in BS (LBS + HBS) treatment decreased by 4.2%, but the number of fungi OTU in BSM (LBSM + HBSM) treatment increased by 14.4%. The OUT-annotation results showed that the number of bacteria and fungi in each treatment changed to varying degrees in terms of kingdom, phylum, class, order, family, genus and species.

Table 1 represents the α diversity analysis of the sample. The Chao1 index of bacteria and fungi in the BS treatment compared with the CK treatment decreased by 14.5% and 5.4%, respectively, while the BSM treatment decreased by 46.8% and 34.5%. This is an indication that BS and BSM treatment reduced the population of species in soil bacterial and fungal communities. Similarly, the Shannon index and Simpson index of bacteria and fungi in BS and BSM treatments were lower than those in CK treatments, indicating that BS and BSM treatments led to a decrease in the diversity of bacteria and fungi communities in soil. Additionally, BS and BSM treatments resulted in the uniformity of species distribution in microbial communities.

#### 2.2.3. Canonical Correspondence Analysis (CCA)

The CCA analysis shows that the content of NH_4_^+^-N in soil was the main contributory factor to the observed inhibition of the growth of bacteria without mulching (Figure 4a,b), while the soil pH was mainly responsible for the inhibition of the growth of bacteria and fungi in soil with mulching (Figure 4c,d). In addition, soil NO_3_^−^-N was positively correlated with the proliferation of bacteria and fungi, and soil OC was positively correlated with the proliferation of fungi, but negatively correlated with the reproduction of bacteria.

### 2.3. Effect of Biogas Slurry Pretreatment on the Production of Capsicum spp.

Figure 5 shows the growth performance of *Capsicum* spp. on biogas slurry-treated soil after 50 days. The survival rate of *Capsicum* spp. pretreated with biogas slurry LBS, HBS, LBSM and HBSM was significantly increased (*p* < 0.05). Comparing the LBS, HBS, LBSM and HBSM with CK treatment, the survival rate of *Capsicum* spp. was increased by 71.3%, 76.2%, 76.2% and 76.2%, respectively. Similarly, comparing these treatments with CKM treatment, an increase of 46.7%, 50.9%, 50.9% and 50.9%, respectively, was obtained. When these treatments were also compared with W treatment, the growth performance increased by 1.2%, 4.1%, 4.1% and 4.1%, respectively (Figure 5A). The rate of rigid seedlings in WM treatment was the highest, reaching 32.5%, followed by W treatment, which was 29.0%. Both treatments were significantly higher than those obtained from CK, CKM, LBS, HBS, LBSM and HBSM (*p* < 0.05). Moreover, comparing CK treatment to LBS, HBS, LBSM and HBSM treatment, the rate of rigid seedlings was observed to reduce by 63.2%, 52.6%, 42.1% and 84.2%, respectively. Similarly, when W treatment was compared with LBS, HBS, LBSM and HBSM treatment, the rate of rigid seedlings was significantly (*p* < 0.05) reduced by 90.4%, 87.7%, 84.9% and 95.9%, respectively (*p* < 0.05) (Figure 5B).

Furthermore, the soil pretreated with biogas slurry significantly increased the plant height of *Capsicum* spp. plants and when this was compared with CK treatment, the plant height of *Capsicum* spp. plants treated with LBS, HBS, LBSM and HBSM increased by 113.1%, 83.6%, 96.7% and 50.8%, respectively (Figure 5C). Additionally, the population of flowering *Capsicum* spp. and the percentage of fruit-bearing *Capsicum* spp. were substantially improved. Comparing CK treatment to other treatments (LBS, HBS, LBSM and HBSM), an increase (98.8%, 82.9%, 86.5% and 81.7%) in the population of flowering *Capsicum* spp. was observed. This was 35.82, 30.08, 31.38 and 29.65 times higher than CK treatments, and 1.64, 1.37, 1.43 and 1.35 times higher than W treatments, respectively (Figure 5D). The highest (47.8%) population of fruit-bearing *Capsicum* spp. was observed with LBS treatment. This was 47.8%, 17.32, 2.31, 19.66 and 5.92 times of CK, W, CKM and WM, respectively (Figure 5E).

## 3. Discussion

### 3.1. Effect of Biogas Slurry Pretreatment on Soil Properties

This study shows that biogas slurry pretreatment of degraded soil improved the physical and chemical properties of the soil [15,16], effectively regulated—the proportion of various nutrients content such as soil organic carbon in the soil [19] and pH value [20]. It has a direct positive effect on soil fertility [18]. Studies have shown that under similar nitrogen conditions, the TN, NH_4_^+^-N, NO_3_^−^-N, AP, AK and pH of soil increased to varying degrees after soil was treated with pig manure biogas slurry to the soil, while the OC content of the soil was not significantly impacted [21]. Similarly, Lai et al. [22], reported controlling the concentration of swine manure biogas slurry applied in three years within the range of 546.25–626.00 × 10^3^ kg/hm^2^ significantly increased soil available potassium, phosphorus, and alkaline hydrolyzable nitrogen, as well as reducing the risk of soil acidification. In another related study, the TN, NH_4_^+^-N, TP, AP, AK and OC content of paddy fields treated with continuous application of biogas slurry for four years were significantly higher than those of no biogas slurry application [23]. The authors also observed an increase in the soil organic matter content which was directly proportional to the amount of biogas slurry applied [24]. The effects of biogas slurry pretreatment on soil TN, TP, AP, AK, and OC in this study were consistent with the results of previous studies. Soil NH_4_^+^-N content decreased in low-volume biogas slurry pretreatment (LBS treatment), which may be related to soil microbial activity, while the relative increase in soil TN was higher in the same treatment. The reduction of NO_3_^−^-N content in the treatments may be ascribed to the downward leaching of NO_3_^−^-N in surface soil caused by irrigation that keeps the soil saturated before pretreatment. The pH value of soil treated with LBS decreased slightly, indicating that low application of biogas slurry could not prevent soil acidification, while high application of biogas slurry fully neutralized soil acids [4], as well as inhibited the growth and proliferation of acidobacteria, thus reducing soil acidification [25].

### 3.2. Effect of Biogas Slurry Pretreatment on Soil Microorganisms

Appropriate application of biogas slurry can promote the growth and enrichment of soil-dominant bacteria and microbial alpha diversity [26,27], and help prevent and control soil-borne diseases of crops [21,28]. Studies showed that the application of biogas slurry could increase the culturable number of soil bacteria [29], fungi [16] and actinomycetes [30] to a certain extent. Although nitrogen input is the key factor in increasing soil microbial nitrogen energy [14], a large amount of ammonium nitrogen in biogas slurry may play a role in inhibiting microbial growth in the short term. Some studies found that the population of bacteria in the soil decreased [9], the main fungi in the soil decreased by 55.03% [31], and *Fusarium oxysporum* decreased significantly [32] after the application of biogas slurry. While with the increase in biogas slurry concentration until the optimal amount, the relative abundance of actinomycetes increases continuously, further increment in the concentration of biogas slurry inhibits the growth of actinomycetes [20].

Furthermore, when the biogas slurry was applied at 180 m^3^/hm^2^, the richness and diversity of bacteria were increased. When the Chao1 index and Shannon index were high, the diversity of fungi decreased with low Chao1 index and Shannon index [25]. In this study, the biogas slurry pretreatment of protected soil showed a reduction in a culturable number of soil microorganisms, a decrease in the richness and diversity of soil microorganisms, and altered the soil-dominant microorganisms. These alterations in the microbial compositions through biogas slurry treatment are not only related to the complex factors of closed greenhouse, high temperature, mulching film, soil moisture and other supporting technologies, but also closely related to the change of soil properties.

### 3.3. Effect of Biogas Slurry Pretreatment on the Production of Crops

Biogas slurry pretreatment measures can promote the growth and development of subsequent crops through fertilization and elimination of soil-borne pathogenic microorganisms, with no risk of affecting crop growth. On the other hand, high NH_4_^+^-N [33], electrical conductivity [34], and chemical oxygen demand [35] in biogas slurry can result in plant roots’ adversity stress, leading to plant growth retardation. In this study, the plant height and flowering rate of *Capsicum* spp. with a low amount of biogas slurry (LBS) were better than those of a high amount of biogas slurry (HBS and HBSM), which buttresses the aforementioned observations. Application of biogas slurry can significantly reduce plant disease index [25]. For instance, a study by Cao et al. [28] showed watermelon *Fusarium* wilt was significantly inhibited, and the disease index was reduced by 36.4%. Moreover, in this study, the survival rates of *Capsicum* spp. with clean water (W, WM) and biogas slurry (LBS, HBS, LBSM, HBSM) were higher than that of the blank control (CK, CKM). This is an indication that soil moisture played an important role in improving the survival rate of *Capsicum* spp. plants. Additionally, the rigid seedling rate of clean water treatment (W, WM) is significantly higher than that of biogas slurry treatment (LBS, HBS, LBSM, HBSM). This substantiates the role of biogas slurry in promoting the growth of crops in biogas slurry pre-treatment soil. This might be related to the nutrient richness of biogas slurry and its potential in the reduction of soil-borne pathogens, but more in-depth research might be needed to substantiate and elucidate this.

### 3.4. Expanding Approaches of Digesting Biogas Slurry in Farmland

The biogas slurry discharged from intensive large-scale livestock and poultry farms, and biogas plants was more than 1.3 billion tons each year in the global area [36,37]. It has exceeded the maximum fertilizer-carrying capacity of plant growth on limited nearby farmland when used as fertilizer, and its application is greatly affected by agricultural seasonality [38]. There are serious secondary pollution environmental risks. It is therefore necessary to explore alternative treatments for more efficient utilization of biogas slurry. This raises the possibility of using biogas slurry in soil pretreatment for continuous cropping in protected land with the aim of achieving a dual purpose of efficient digestion of biogas slurry per unit area of farmland, and prevention and control of soil-borne diseases in protected land. It is different from the traditional approaches such as seed soaking, foliar fertilizer, base fertilizer, top dressing fertilizer, hydroponics and animal feed [39,40,41]. To ensure the environmental safety of this approach, the soil maximum adsorption on biogas slurry ammonium nitrogen was researched [42], and the aging characteristics and fate analysis of biogas slurry ammonium nitrogen disposal in soil were also researched by simulation experiments [43] before this study.

## 4. Materials and Methods

### 4.1. Experiment Site and Soil Properties

The experiment was carried out in a *Capsicum* spp. (variety was Moxiu No. 8) greenhouse with a latitude of N:33°10′47.51″ and longitude of E:118°09′4.71″ in Sihe Township, Sihong County, China. *Capsicum* spp. was planted continuously for more than 5 years in the greenhouse. The basic soil properties of the greenhouse after the harvest of the previous crop were organic carbon (OC) 8.92 ± 0.61 g/kg, pH 7.40 ± 0.02, total nitrogen (TN) 1.49 ± 0.15 g/kg, ammonium nitrogen (NH_4_^+^-N) 49.37 ± 1.67 mg/kg, nitrate nitrogen (NO_3_^−^-N) 351.87 ± 13.81 mg/kg, total phosphorus (TP) 1.09 ± 0.02 g/kg, available phosphorus (AP) 21.15 ± 2.78 mg/kg and available potassium (AK) 242.35 ± 11.12 mg/kg. Soil culturable microbial indicators: bacteria (3.20 ± 0.19) × 10^6^ cfu/g, fungi (1.00 ± 0.06) × 10^4^ cfu/g, actinomycetes (2.84 ± 0.09) × 10^6^ cfu/g, genus *Fusarium* (1.08 ± 0.14) × 10^3^ cfu/g.

### 4.2. Biogas Slurry Properties

The biogas slurry used in this study was obtained from a biogas plant in the adjacent pig farm, which was produced by pig excreta and wastewater through high performance half-underground anaerobic digester with continuous process. The volume of the anaerobic fermentation tank is 1960 m^3^, and the biogas slurry storage tank is 1800 m^3^. The excreta and wastewater flow from the inside of the pig farm through pipelines to a mixing tank, and then is continuously pumped into the anaerobic fermentation tank. After 45 to 60 days of fermentation, biogas slurry automatically flows out under water pressure and enters the storage tank. The average properties of biogas slurry are TN 628.34 ± 103.22 mg/L, NH_4_^+^-N 519.89 ± 96.83 mg/L, NO_3_^−^-N 65.64 ± 8.58 mg/L, TP 339.72 ± 96.13 mg/L, total potassium (TK) 423.47 ± 81.67 mg/L, COD 1060 ± 35 mg/L and pH 8.14 ± 0.31.

### 4.3. Design and Setting

A split-plot experimental design was adopted in the application of the biogas slurry. Four main farmland plots were implemented: the blank control without any treatment (CK); the clean water treatment (W) (applying clean water of 495 m^3^/hm^2^), the low irrigation volume biogas slurry treatment (LBS) (applying biogas slurry of 495 m^3^/hm^2^) and the high irrigation volume treatment (HBS) (applying 990 m^3^/hm^2^ of biogas slurry). The W, LBS and HBS treatments were brought to soil-saturated water content by watering 450 m^3^/hm^2^ a day prior to applying the biogas slurry on the farmland. Two additional subplots were set up, one with the black mulch and another without the black mulch. Thus, a total of eight treatments, namely CK, W, LBS, HBS, CKM, WM, LBSM and HBSM, were implemented. Each experimental treatment was repeated three times. The area of each plot is 35 m^2^ (length 7 m, width 5 m), with an underground depth of 80 cm, and a reserved height above ground of 20 cm with a black PE impermeable membrane buried around the plot. The biogas slurry was extracted by a mud pump and then applied using a flow meter.

After the application of biogas slurry, the facility greenhouse was closed for 20 days, then the covering film was removed, and the greenhouse was further kept ventilated for 7 days, after which the land was plowed and aired for another 14 days. The *Capsicum* spp. seedlings were transplanted after a 51-day seedling period using local farmer transplanting density. During the *Capsicum* spp. planting, 45% of fertilizer was used as base fertilizer, 30% as topdressing at the flowering stage and 25% as topdressing at the fruit stage. The daily management of pest control, weeding and water irrigation was carried out using conventional methods.

### 4.4. Sampling and Analysis

#### 4.4.1. Determination Content

Soil samples were taken at 5, 10, 15 and 20 days after biogas slurry application to determine the soil properties. The soil properties analyzed were TN, NH_4_^+^-N, NO_3_^−^-N, TP, AP, AK, OC, pH, soil-culturable bacteria, fungi, actinomycetes and fusarium population. Additionally, microbial community diversity using 16S rDNA V3 + V4 area/fungal ITS rDNA ITS2 area sequencing was carried out.

The growth of *Capsicum* spp. in each treatment was measured 50 days after transplanting. Data on the total number of transplanted *Capsicum* spp., seedlings survival rate, rate of rigid seedlings, rate of flowering plants, rate of fruit-bearing plants and plant height were obtained using Equations (1)–(4).
(1)$$Survival\;rate\;seedlings\;\left(\%\right)=\frac{Survival\;seedling\;number}{Total\;number\;of\;transplanted}\times 100$$
(2)$$Rigid\;seedlings\;rate\;\left(\%\right)=\frac{Hardened\;seedlings\;number}{Survival\;seedlings\;number}\times 100,$$
(3)$$Flowering\;plants\;rate\;\left(\%\right)=\frac{Flowering\;plants\;number}{Survival\;seedlings\;number}\times 100,$$
(4)$$Fruit-bearings\;rate\;\left(\%\right)=\frac{Fruit-bearing\;plants\;number}{Survival\;seedlings\;number}\times 100$$

#### 4.4.2. Soil Sample Collection Methods

The multi-point sampling method was used for the soil sampling analysis. The soil was collected from a 0–10 cm soil layer with an undisturbed soil drill. The fresh soil sample was spread on clean paper and mixed evenly. Then, 30 g of finely crushed fresh soil was stored in a ziplock PE bag at 4 °C and thereafter used to enumerate microorganisms present in the soil. Another 5 g of fine fresh soil was wrapped in an aluminum foil and then stored in liquid nitrogen for the determination of microbial community diversity. The remaining soil sample was dried, sieved and used for the determination of soil properties.

#### 4.4.3. Soil Properties Determination

The chemical properties of the soil were determined using standard methods for chemical analysis of soil and agriculture [44]. First, the soil sample was digested with concentrated sulfuric acid and the content of TN and TP was determined with an automatic flow analyzer (SKALAR SAN^++^, SKALAR, Breda, The Netherlands). Then, the contents of NH_4_^+^-N and NO_3_^−^-N in the soil were extracted with 2 mol/L KCl solution (ratio of 2 mol/L KCl solution to the soil was 5:1), and determined with SKALAR SAN^++^ (Holland) analyzer. Moreover, AP content was obtained by sodium bicarbonate leaching–molybdenum antimony anti-spectrophotometry (HJ 704-2014), while AK content was determined by ammonium acetate extraction flame photometer method (NY/T 889-2004) [45]. The OC was determined using the potassium dichromate volumetric method [46]. Lastly, the pH value was measured by Thunder Magnetic PHS-3C pH meter (INESA, Shanghai, China) [47,48].

#### 4.4.4. Soil Microbial Community Determination

The gradient dilution plate coating counting method was used to determine the culturable bacteria, fungi, actinomycetes and *Fusarium*. The beef extract peptone agar medium (NA bacterial medium, with the composition of beef paste 3 g, peptone 10 g, NaCl 5 g, agar 20 g, 1 L deionized water, pH = 7.4–7.6, sterilized at 121 °C for 30 min) was employed for the culturable bacteria, the Martin’s medium (with the composition of glucose 10 g, peptone 5 g, MgSO_4_·7H_2_O 0.5 g, KH_2_PO_4_ 1 g, agar 20 g, bengal red 0.03 g, streptomycin 0.03 g, deionized water 1 L) was employed for the culturable fungi, the modified Gao’s No.1 medium (actinobacteria culture medium, with the composition of soluble starch 20 g, KNO_3_ 1 g, K_2_HPO_4_ 0.5 g, MgSO_4_·7H_2_O 0.5 g, NaCl 0.5 g, FeSO_4_·7H_2_O 0.01 g, agar 20 g, pH = 7.4–7.6, add 10% phenol 2 drops to 100 mL of culture medium before pouring the dish and mix well) was employed for the culturable actinomycetes, and the modified Komada’s selective medium (*Fusarium* specific selection medium, with the composition of Na₂B₄O₇ 1 g, K_2_HPO_4_ 1 g, KCl 0.5 g, MgSO_4_·7H_2_O 0.5 g, EDTA sodium iron salt 0.01 g, D-galactose 20 g, L-asparagine 2 g, agar 15 g, pentachloronitrobenzene (PCNB)(75%) 1 g, bovine bile salt 0.5 g, streptomycin sulfate 0.3 g, distilled water 1 L, pH 4.0) was employed for the culturable *Fusarium* [49]. This was achieved by weighing 10 g of soil into 90 mL of sterile water and incubating for 1 h at 28 °C constant temperature oscillation. Then, it was diluted according to a 10 times gradient and coated with a specific culture medium at corresponding dilution times (coating was carried out at 330 uL per plate). This was repeated three times, and enumerated after 2–5 days of incubation [50].

Soil microbial community diversity was determined using high-throughput sequencing. The specific steps are as follows:

① DNA extraction and quality control: The FastDNA^®^ SPIN Kit (MPBIO, Santa Ana, CA, USA) was used to extract the total microbial DNA in the soil [51]. The specific experimental steps were carried out according to the manufacturing instructions; 1% agarose gel electrophoresis was used to detect the degradation and impurities of the extracted DNA samples. NanoDrop nucleic acid protein analyzer (Thermo Fisher Scientific, Waltham, MA, USA) was used to detect the sample concentration and total amount of DNA, and the PCR pre-amplification to test whether the sample was qualified.

② 16S/ITS rDNA library preparation: Ten nanograms (10 ng) of DNA template was used to carry out PCR amplification of the target region; bacterial 16S rDNA V3 + V4 region amplification primers were 338F (5′-ACTCCTACGGGAGGCAGCA-3′) and 806R (5′-GGACTACHVGGGTWTCTAAT-3′), fungal ITS rDNA ITS2 region amplification primers are ITS3F (5′-GCATCGATGAAGAACGCAGC-3′) and ITS4R (5′-TCCTCCGCTTATTGATATGC-3′) [25,52]. The PCR amplification was divided into two steps: first, specific primers to amplify the target fragment with the EX Taq enzyme (TaKaRa, Osaka, Japan) to ensure amplification efficiency and accuracy. The target fragment was purified and recovered by Novozen AMPure XP magnetic beads (Beckman Coulter, Brea, CA, USA), and then the recovered product was used as a template for secondary PCR amplification. The connectors, sequencing primers and barcodes required for sequencing on the Illumina platform were added to both ends of the target fragment. After the library was constructed using a Qubit 2.0 Fluorometer (Thermo Fisher Scientific, MA, USA) for preliminary quantification and diluting the library to 1 ng/μL, Agilent2100 (Agilent, Santa Clara, CA, USA) was used to detect the insert size of the library.

③ Computer sequencing: Paired-end sequencing was performed on the Illumina Hiseq platform for qualified libraries using the PE250 sequencing strategy.

④ Sequencing analysis: The data filtering is completed by removing low-quality bases, Ns and linker contamination sequences to obtain credible target sequences for subsequent analysis.

### 4.5. Data Analysis

Office Excel 2016 software and OriginPro 2017 were used to analyze and map soil traits, soil microorganisms and *Capsicum* spp. plant growth data. One-way ANOVA and Duncan’s method were further used for the analysis of variance and multiple comparisons (α = 0.05) using IBM SPSS Statistics (version 22).

## 5. Conclusions

This study has demonstrated the potential of biogas slurry pretreatment of protected soil to improve soil fertility, alleviating continuous cropping obstacles and promoting plant growth for high productivity. Soil pretreatment with biogas slurry dosage of 495 m^3^/hm^2^ and 990 m^3^/hm^2^ increased soil total nitrogen, total phosphorus, available phosphorus, available potassium, organic carbon and soil pH, while inhibiting the growth of soil bacteria, fungi, actinomycetes and *Fusarium*. It also reduced the soil microbial flora as well as the evenness of species distribution. Moreover, the soil ammonium nitrogen, soil pH and soil nitrate nitrogen were closely correlated to the growth and proliferation of soil bacteria and fungi. Interestingly, a biogas slurry dosage of 495 m^3^/hm^2^ improved the growth of *Capsicum* spp., which significantly improved the survival rate of *Capsicum* spp. seedlings, the plant height, the flowering rate and the fruit-bearing rate. In addition, the rate of rigid seedlings was substantially reduced. These findings demonstrate that biogas slurry, which is usually disposed of in unfriendly ways to the environment, can be an excellent sustainable source for improving soil fertility and crop productivity.

## Figures and Tables

**Figure 1 plants-13-00265-f001:**
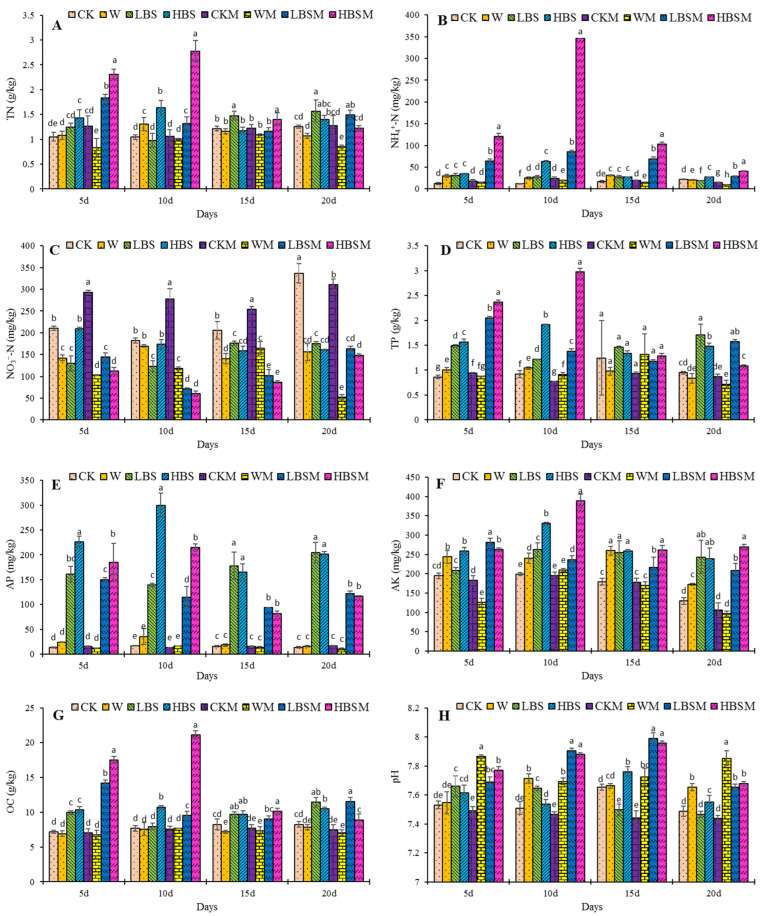
Changing trend of soil properties following biogas slurry pretreatment. (**A**): TN (total nitrogen); (**B**): NH_4_^+^-N (ammonium nitrogen); (**C**): NO_3_^–^-N (nitrate nitrogen); (**D**): TP (total phosphorus); (**E**): AP (available phosphorus); (**F**): AK (available potassium); (**G**): OC (organic carbon); (**H**): pH. The lowercase letters a, b, c, d, e, f, g and h in the figure show observable significant differences (*p* < 0.05).

**Figure 2 plants-13-00265-f002:**
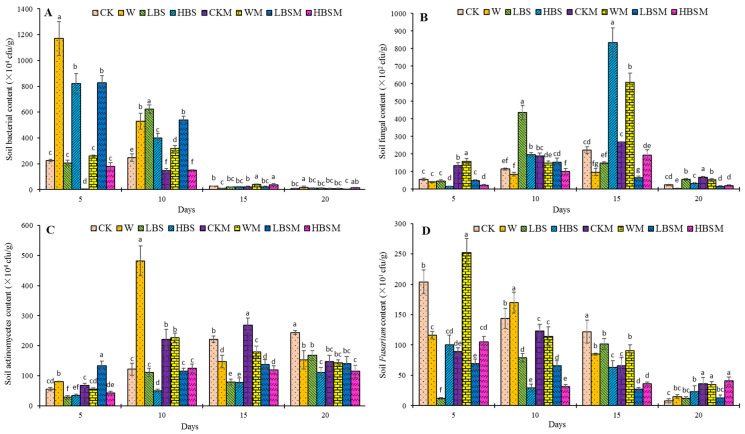
Profile of culturable bacteria, fungi, actinomycetes and *Fusarium* in biogas slurry pretreated soil. (**A**): bacteria; (**B**): fungi; (**C**): actinomycetes; (**D**): *Fusarium*. The lowercase letters a, b, c, d, e, f and g in the figure show observable significant differences (*p* < 0.05).

**Figure 3 plants-13-00265-f003:**
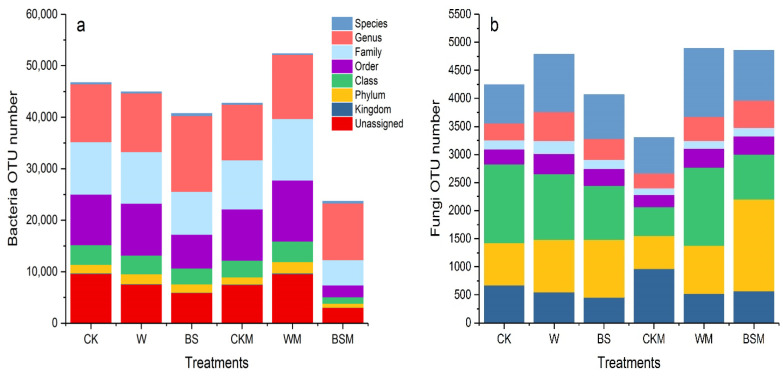
OTU cluster analysis and annotation results of soil bacteria and fungi in pretreated soil. (**a**): bacteria OUT number; (**b**): fungi OUT number.

**Figure 4 plants-13-00265-f004:**
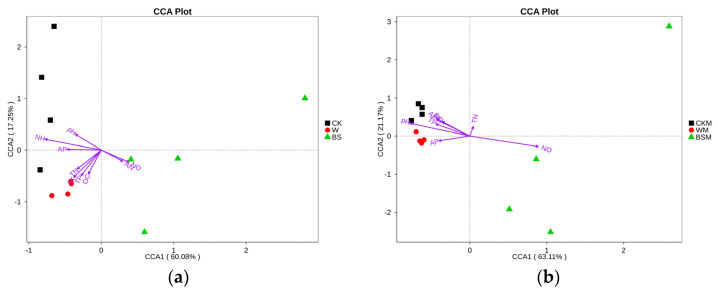
Canonical correspondence analysis (CCA) between soil bacteria (**a**,**b**), fungi (**c**,**d**) and soil properties.

**Figure 5 plants-13-00265-f005:**
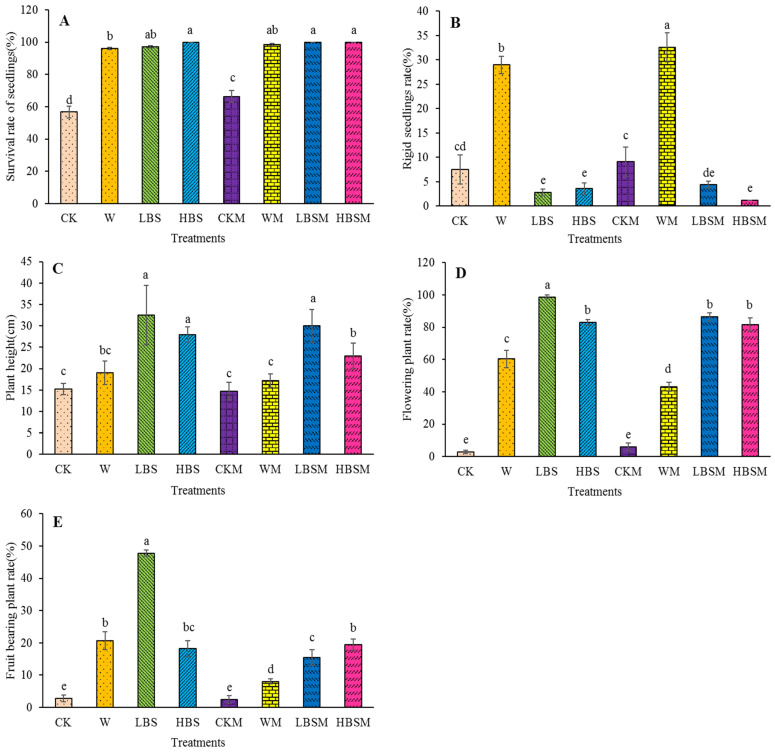
Growth of *Capsicum* spp. under different biogas slurry pretreatments. (**A**): survival rate of seedlings; (**B**): rigid seedlings rate; (**C**): plant height; (**D**): flowering plant rate; (**E**): fruit-bearing plant rate. The lowercase letters a, b, c, d and e in the figure show observable significant differences (*p* < 0.05).

**Table 1 plants-13-00265-t001:** Change in α diversity indices of bacteria and fungi under different treatments.

Type	Sample Name	Shannon	Simpson	Chao1	Goods Coverage
Bacteria	CK	11.14 ± 0.61 ab	0.99 ± 0.00 a	19,238.07 ± 5150.93 ab	0.93 ± 0.03 ab
	W	11.44 ± 0.34 ab	1.00 ± 0.00 a	19,913.37 ± 5801.58 ab	0.93 ± 0.03 ab
	BS	10.51 ± 0.27 b	0.99 ± 0.00 a	16,453.05 ± 2047.63 ab	0.94 ± 0.01 a
	CKM	11.30 ± 0.73 ab	1.00 ± 0.00 a	31,899.50 ± 23,013.83 a	0.88 ± 0.05 bc
	WM	11.91 ± 0.28 a	1.00 ± 0.00 a	27,582.11 ± 7239.36 a	0.87 ± 0.03 c
Fungi	CK	6.59 ± 0.14 a	0.97 ± 0.00 a	1956.38 ± 514.75 ab	0.99 ± 0.00 a
	W	6.09 ± 0.24 a	0.95 ± 0.02 a	1630.31 ± 387.41 b	1.00 ± 0.00 a
	BS	6.17 ± 0.85 a	0.96 ± 0.02 a	1851.14 ± 606.25 ab	1.00 ± 0.00 a
	CKM	6.54 ± 0.16 a	0.97 ± 0.00 a	2494.80 ± 756.80 a	0.99 ± 0.00 a
	WM	6.23 ± 0.43 a	0.95 ± 0.02 a	1868.95 ± 146.70 ab	1.00 ± 0.00 a
	BSM	3.45 ± 2.82 b	0.55 ± 0.39 b	1281.76 ± 118.88 b	1.00 ± 0.00 a

The lowercase letters a b and c followed the number in the table show observable significant difference (*p* < 0.05).

## Data Availability

Data will be made available on request.

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
