# Peer review of "Biogas Slurry Significantly Improved Degraded Farmland Soil Quality and Promoted Capsicum spp. Production"

_plants, 2024, doi:10.3390/plants13020265_

Round 1
Reviewer 1 Report
Comments and Suggestions for Authors
Author Response
We greatly appreciate your affirmation of our research work. Your positive comments and insightful suggestions inspired us to take further research in this field.
Point 1: “2. Results:
Just a few notes to change: In lines 187, 189 and 195, like the other sub-chapters, it is important to put the figures (4 and 5) in bold.
The text could be further divided into more paragraphs. Very long prose creates some difficulty in reading.”
Response 1: Revised as suggested. We divided divided “2.1. Effect of biogas slurry pretreatment on soil properties…” into three paragraphs, divided “2.2.1. Soil culturable microorganisms…” into three paragraphs, divided “2.2.2. Changes of soil microbial community diversity…” into two paragraphs, and divided “2.3. Effect of biogas slurry pretreatment on the prodution of Capsicum spp.…” into two paragraphs.
Point 2: “4. Materials and Methods:
However I point out some references that you can correct:
4.2. Biogas slurry properties. The last phrase (line 330) “The experimental Capsicum spp. variety was Moxiu No. 8” you can go to point 4.1 right at the beginning when you introduce the species Capsicum spp, and delete that sentence in 4.2 sub-chapter.
For example:
4.1. Experiment site and soil properties
The experiment was carried out in a Capsicum spp. (variety Moxiu No. 8”)”
Response 2: Revised as suggested.
Point 3: “References:
The work cites 52 references and are very diverse (years and authors). However, reference 40 is missing in the text.”
Response 3: Revised as suggested. Lines 300-302 “It is different from the traditional approaches such as seed soaking, foliar fertilizer, base fertilizer, top dressing fertilizer, hydroponics and animal feed[39-41] ”.
Point 4: “GENERAL INFORMATION:
Pay attention to the names in Latin: of the species you are working on as well as the fungi. All these names must be in italics.
Another important aspect is the presentation of the document. Always leave similar spaces between chapters and sub-chapters.
Pay attention to the words in bold.Pay attention to the size of the paragraphs.
The article is very well written, very clear and is very useful for the international community.”
Response 4: Revised as suggested.

Reviewer 2 Report
Comments and Suggestions for Authors
Revision / Manuscript ID: 2760998
The figures in this Manuscript are unreadable.
The Abstract give the picture of the entire manuscript.
Introduction: The manuscript's content is of local interest but the Authors did not explain while the study is of broad international interest.
The 'Results' is very important part of the manuscript. It needs to be clear. Quality of the figures 1 and 2 must be enhanced they are unreadable (mainly see the patterns and letters) in the figures 4 and 5 (see the scales).
Materials and methods:
Please specify the composition of the media and their modifications.
Provide information about which group of microorganisms are identified on each type of medium.
Correct the spelling of species names and genus names (for italic).
See also: L: 77 “Oxysporum” -what is it?
Explain the term “actinomycetes” – Authors divide microorganisms for: bacteria and actinomycetes (actinomycetes are group of Gram positive bacteria) it should be clarified at the beginning of the Manuscript.
Explain what is “fusarium” in this Manuscript? (I know the genus Fusarium and species Fusarium sp.).

Author Response
Response to Reviewer 2
Comments 1: The figures in this Manuscript are unreadable.
The 'Results' is very important part of the manuscript. It needs to be clear. Quality of the figures 1 and 2 must be enhanced they are unreadable (mainly see the patterns and letters) in the figures 4 and 5 (see the scales).
Response 1: Thank you for your suggestion. We have changed the “figures” into “editable data figures”.
Comments 2: Introduction: The manuscript's content is of local interest but the Authors did not explain while the study is of broad international interest.
Response 2: Revised as suggested. Added the word “global” (L47). Changed the “Efforts have been made to” into “Scholars from Japan, the Netherlands, the United States, and China have made efforts to” (L59-60). At the end of the Discussion chapter (3.4. Expanding approaches of digesting biogas slurry in farmland), we discussed the purpose of this study which will be of broad international interest in the futuer.
Comments 3: Materials and methods:
Please specify the composition of the media and their modifications.
Provide information about which group of microorganisms are identified on each type of medium.
Response 3: Revised as below:
The gradient dilution plate coating counting method was used to determine the culturable bacteria, fungi, Actinomycetes, and Fusarium. The beef extract peptone agar medium (NA bacterial medium, with the composition of beef paste 3g, peptone 10g, NaCl 5g, agar 20g, 1L deionized water, pH=7.4-7.6, sterilized at 121 ℃ for 30 minutes) was employed for the culturable bacteria, the Martin's medium (with the composition of glucose 10g, peptone 5g, MgSO4·7H2O 0.5g, KH2PO4 1g, agar 20g, bengal red 0.03g, streptomycin 0.03g, deionized water 1L) was employed for the culturable fungi, the modified Gao's No.1 medium (Actinobacteria culture medium, with the composition of soluble starch 20g, KNO3 1g, K2HPO4 0.5g, MgSO4·7H2O 0.5g, NaCl 0.5g, FeSO4·7H2O 0.01g, agar 20g, pH=7.4-7.6, add 10% phenol 2 drops to 100mL of culture medium before pouring the dish and mix well) was employed for the culturable Actinomycetes, and the modified Komada's selective medium (Fusarium specific selection medium, with the composition of Naâ‚‚Bâ‚„O₇ 1g, K2HPO4 1g, KCl 0.5g, MgSO4·7H2O 0.5g, EDTA sodium iron salt 0.01g, D-galactose 20g, L-asparagine 2g, agar 15g, pentachloronitrobenzene (PCNB)(75%) 1g, bovine bile salt 0.5g, streptomycin sulfate 0.3g, distilled water 1L, pH 4.0) was employed for the culturable Fusarium.
Comments 4: Correct the spelling of species names and genus names (for italic).
See also: L: 77 “Oxysporum” -what is it?
Explain the term “actinomycetes” – Authors divide microorganisms for: bacteria and actinomycetes (actinomycetes are group of Gram positive bacteria) it should be clarified at the beginning of the Manuscript.
Explain what is “fusarium” in this Manuscript? (I know the genus Fusarium and species Fusarium sp.).
Response 4: We greatly appreciate your insightful suggestions. Revised as follows:
Comments: Correct the spelling of species names and genus names (for italic)
Response: Revised as suggested.
Comments: See also: L: 77 “Oxysporum” -what is it?
Response: We have revised the “Oxysporum” into “Fusarium oxysporum” (in italic).
Comments: Explain the term “actinomycetes” – Authors divide microorganisms for: bacteria and actinomycetes (actinomycetes are group of Gram positive bacteria) it should be clarified at the beginning of the Manuscript.
Response: Revised as suggested (2.2.1. Soil culturable microorganisms, L131).
Comments: Explain what is “fusarium” in this Manuscript? (I know the genus Fusarium and species Fusarium sp.).
Response: We have revised the “fusarium” into “genus Fusarium (all culturable Fusarium species, including the pathogenic fungus causing Capsicum spp. root rot)”(2.2.1. Soil culturable microorganisms, L132-133).

Round 2
Reviewer 2 Report
Comments and Suggestions for Authors
Revision 2/ Manuscript ID: 2760998
The 'Results' is very important part of the manuscript.
It’s hard to read this Manuscript without Figures 1 and 2.
Quality of the figures 1 and 2 must be enhanced they are unreadable (mainly see the patterns and letters).
-Why the Authors did not use the colors or
- Figures 1 and 2 should be as large as the Figures 5 are.
- See the Figures 3 which are PERFECT.
Again, please clarify “Actinomycetes” why is it in ITALIC ?
Please correct it in all the Manuscript.
There is a difference when You are writing (Actinomycetes, actinomycetes) – for a nontaxonomic term, for a group of common soil microorganisms and You can also use Actinomycetes spp.

Author Response
Comments 1:
The 'Results' is very important part of the manuscript.
It’s hard to read this Manuscript without Figures 1 and 2.
Quality of the figures 1 and 2 must be enhanced they are unreadable (mainly see the patterns and letters).
-Why the Authors did not use the colors or
- Figures 1 and 2 should be as large as the Figures 5 are.
- See the Figures 3 which are PERFECT.
Response 1: We greatly appreciate your insightful suggestions. We have changed the “figures” 1, 2 and 5 using the colors.
Comments 2:
Again, please clarify “Actinomycetes” why is it in ITALIC ?
Please correct it in all the Manuscript.
There is a difference when You are writing (Actinomycetes, actinomycetes) – for a nontaxonomic term, for a group of common soil microorganisms and You can also use Actinomycetes spp.
Response 2: Thank you for your suggestion again. We have revised the “Actinomycetes” into “actinomycetes” in all the Manuscript. And we have added “actinomycetes (group of Gram positive bacteria, different from bacteria in terms of cellular characteristics and metabolic functions, actinomycetes can produce various secondary metabolites such as antibiotics and play an important role in controlling harmful micro-organisms in soil)” (2.2.1. Soil culturable microorganisms, L133-136).

Round 3
Reviewer 2 Report
Comments and Suggestions for Authors
I would like to thank the Authors for response to all received comments.and I accept the Manuscript (Plants-2760998) in the present form